# Promises and Challenges of Immunogenic Chemotherapy in Multiple Myeloma

**DOI:** 10.3390/cells11162519

**Published:** 2022-08-14

**Authors:** Megan Johnstone, Delaney Vinaixa, Marcello Turi, Eugenio Morelli, Kenneth Carl Anderson, Annamaria Gulla

**Affiliations:** 1Jerome Lipper Multiple Myeloma Center, LeBow Institute for Myeloma Therapeutics, Dana-Farber Cancer Institute, Boston, MA 02215, USA; 2Faculty of Science, University of Ostrava, 70100 Ostrava, Czech Republic; 3Faculty of Medicine, University of Ostrava, 70300 Ostrava, Czech Republic; 4Department of Hematooncology, University Hospital Ostrava, 70800 Ostrava, Czech Republic; 5Harvard Medical School, Boston, MA 02115, USA

**Keywords:** myeloma, ICD, immunogenic chemotherapy, DAMPs, microenvironment

## Abstract

Immunological tolerance of myeloma cells represents a critical obstacle in achieving long-term disease-free survival for multiple myeloma (MM) patients. Over the past two decades, remarkable preclinical efforts to understand MM biology have led to the clinical approval of several targeted and immunotherapeutic agents. Among them, it is now clear that chemotherapy can also make cancer cells “visible” to the immune system and thus reactivate anti-tumor immunity. This knowledge represents an important resource in the treatment paradigm of MM, whereas immune dysfunction constitutes a clear obstacle to the cure of the disease. In this review, we highlight the importance of defining the immunological effects of chemotherapy in MM with the goal of enhancing the clinical management of patients. This area of investigation will open new avenues of research to identify novel immunogenic anti-MM agents and inform the optimal integration of chemotherapy with immunotherapy.

## 1. Introduction

Over the past several decades, the treatment paradigm of multiple myeloma (MM) has dramatically evolved to improve patient outcomes. Patients are currently treated using a combination of MM-targeted agents and immunotherapy to not only directly kill neoplastic plasma cells but also indirectly kill them by activating relevant components of the immune system [1,2]. However, the onset of drug resistance and tumor immune escape eventually lead to disease relapse and therapeutic failure [1].

As most anti-MM agents were initially tested in immunodeficient mice, the contribution of the immune system to their therapeutic efficacy has historically been understudied [3]. Interestingly, it is now clear that several chemotherapies regulate the immune system through tumor-intrinsic or extrinsic mechanisms. This point is extremely relevant in the treatment scenario of MM, whereas the long-term clinical success of immunotherapy is hampered by the presence of an immunosuppressive bone marrow milieu and exhausted or dysfunctional T lymphocytes [1,2]. Among the tumor-intrinsic mechanisms of chemotherapy induced immunomodulation stands the relatively new concept of “immunogenic cell death” (ICD), whereby a cancer cell dies in a way that stimulates innate and/or adaptive immunity. This concept offers clear opportunities to define the ideal combination of chemo- and immunotherapeutic approaches to inform clinical practice.

In this review, we will summarize the present knowledge on the mechanisms underlying ICD and how we can exploit it in the current MM treatment landscape. Furthermore, we will discuss the challenges of immune evasion and ICD resistance as well as their relevance in the scope of MM biology. We anticipate that the integration of such knowledge will improve chemo- and immunotherapy combinations that will impact MM treatment and improve the long-term survival of MM patients. 

## 2. Immunogenic Cell Death: Using Cancer to Beat Cancer

Until recently, it was believed that chemotherapies worked via mechanisms that were independent of the immune system. While this remains partially true, recent discoveries have shown that certain chemotherapies can also induce an immune response that augments their therapeutic success. Dependent on the initiating stimulus, cell death can be immunogenic. Such immunogenic cell death (ICD) describes a change in cell surface composition and the release of soluble mediators, occurring in a defined spatiotemporal sequence in dying cancer cells [4]. These mediators act as danger signals, and allow the immune system to essentially use the cancer cells to its benefit and mount a more effective anti-tumor response [4]. 

### 2.1. Immunogenic Danger Signals from Dying Cancer Cells

During treatment related induction of ICD, cancer cells express or release damage-associated molecular patterns (DAMPs) that stimulate an anti-cancer immune response by interacting with pattern recognition receptors (PRRs) on immune cells [5]. These DAMPs constitute a cellular “danger state” and act as adjuvant signals to promote co-stimulation and pro-inflammatory molecules to foster T cell priming by APCs and adaptive immunity [6,7,8]. The successful induction of ICD is defined by the quality of the cellular dialog between dying cancer cells that emit the DAMPs and the immune cells that perceive these immunogenic signals [6,7,9]. 

DAMPs exposure and/or release is the result of stress-responsive molecular pathways underlying ICD including (1) endoplasmic reticulum (ER) stress and the unfolded protein response (UPR), (2) autophagy, (3) a viral mimicry state and the type-I interferon (IFN) response, as well as (4) the release of “hidden” molecules resulting from the loss of plasma membrane integrity (Figure 1). Here, we will describe the different signaling and key molecules that are relevant within the scope of MM biology. As detailed below, there are multiple points at which tumor cells or therapeutic agents can amplify or minimize the impact of ICD.

#### 2.1.1. ER Stress Response and Calreticulin Exposure 

MM cells are characterized by protein overload due to high protein turnover and high paraprotein production [1]. Therefore, they are particularly susceptible to therapeutic interventions, such as proteasome inhibitors like bortezomib and carfilzomib, that disequilibrate protein homeostasis [5,10]. These inhibitors cause the accumulation of misfolded proteins that triggers the ER stress response, which in turn triggers both the integrated stress response (ISR) [11] and the UPR. The ISR pathway involves the activation of eukaryotic translation initiation factor 2 subunit alpha (eIF2α) via phosphorylation, a hallmark of effective ICD induction. Across a variety of cancers, eIF2α phosphorylation is positively correlated with calreticulin (CRT) exposure, higher tumor immune infiltration (by dendritic cells (DCs) and cytotoxic T lymphocytes (CTLs)) and better clinical outcome in patients [6,7,12,13]. 

CRT is an abundant ER chaperone that translocates from the ER to the surface of the plasma membrane in dying cancer cells exclusively following treatment with ICD inducers [12,13]. CRT exposure occurs as early as the pre-apoptotic stage of cell death and persists in enucleated cells, thus occurring independently from purely nuclear components, the DNA damage response or transcriptional reprogramming [3,4,14]. This crucial step, which is not a general feature of cell death, has proved obligatory for the induction of ICD, as CRT functions as a prominent “eat-me” signal and drives the engulfment of dying cancer cells by DCs via binding to the transmembrane receptor CD91 (also known as LRP1) on DCs and macrophages [3,14]. Loss of CRT surface expression (or any of the proteins involved in its translocation) abolishes the in vitro and in vivo immunogenicity of dying cancer cells, thereby reducing the overall anti-tumor immune response [4,14]. 

Early exposure of CRT may be followed by the exposure of other molecular chaperones, such as heat-shock protein 90 (HSP90), which similarly facilitates tumor cell-DC engulfment and DC maturation [3]. Though the net contribution of the different chaperones to overall immunogenicity appears dependent on the stimulus and cell context, evidence so far has only formally proved the obligate role of CRT in the ICD process [14]. 

In the context of MM, bortezomib and the B-cell maturation antigen (BCMA)-targeting antibody–drug conjugate belantamab mafodotin trigger ICD through surface expression of both HSP90 and CRT [5,15,16]. Similarly, epidrugs such as DNA methyltransferase inhibitor decitabine and the histone deacetylase inhibitor quisinostat have shown to mediate CALR exposure and ICD [17]. Thus far, there is only in vivo evidence to show the obligate contribution of CRT to the immunogenicity of MM cells [5]. Though many therapeutic agents for MM disequilibrate the proteasome and activate the ER stress and the ISR pathways, bona fide ICD still requires additional biological markers to be activated in vivo [18]. That being said, predictions based on the activity of drugs in MM cells may still be helpful for identifying ICD inducers.

#### 2.1.2. Autophagy and ATP Release 

A central component of the ISR pathway is the stimulation of autophagy. Autophagy can promote innate and adaptive immunity by regulating antigen sequestration, accumulation, and degradation, as well as the release of DAMPs and the adjuvanticity of tumor cells [6,7,19,20]. Indeed, depending on the context, autophagy can evoke immunostimulatory effects across different stages of tumor progression and therapeutic regimens [19,21,22,23]. 

In the context of chemotherapy-induced ICD, autophagy is necessary for the release of adenosine trisphosphate (ATP), which constitutes an important extracellular DAMP [6,24,25]. This potent “find me” signal binds to ionotropic or metabotropic purinergic receptors to recruit immune cells, specifically DCs and macrophage precursors, to the tumor to aid in their differentiation and subsequent priming of antitumor T cell immunity [4,6,25,26,27]. ATP binding to the purinergic receptor P2X 7 (P2RX7) receptor can additionally mediate the activation of the NLRP3 inflammasome and secretion of interleukin-1 beta (IL-1β), which primes tumor-specific IFN-γ-producing T cells [25,28,29]. Disruption at any point of the autophagy cascade, such as the sequestration of cytoplasmic material, fusion of autophagosomes with lysosomes, and degradation of autophagosomal cargo by lysosomal hydrolases, limits ATP release from dying cells and hampers an effective adaptive immune response [4,25]. Accordingly, deficiency in the autophagy pathway mediates tumor immune escape, low immune infiltration of CTLs and poor prognosis in breast cancer [19,30]. In line with this observation, stimulation of autophagy by dietary regimens (such as time-restricted fasting) and fasting-mimetic agents potentiate the effects of ICD inducers and immunotherapy, including programmed cell death protein 1 (PD-1) blockade, in several preclinical cancer models [6,7,31,32]. Additionally, it’s important to note that inducing autophagy in immune cells, as opposed to cancer cells, may also affect immune competence [6,7]. Autophagy is dynamically regulated during the CD8^+^ T cell response and may support CD8^+^ T cell transition into memory T cells by regulating T cell metabolism [33].

Autophagy plays a key role in MM cell biology, as it cooperates with the proteasome system to degrade misfolded proteins [34]. For this reason, inhibition as well activation of autophagy over a significant threshold, induces MM cell death. Several anti-MM agents, including BTZ and Belantamab Mafodotin, activate the autophagy pathway during cytotoxic stress and are associated with ATP release during ICD [16]. Therefore, future studies are needed to explore the contribution of autophagy induced by anti-MM agents in the context of immunogenicity.

#### 2.1.3. Viral Mimicry

A chemotherapy-induced immune response may also mimic those induced by viral infection, whereby a cancer cell will foster a pro-inflammatory microenvironment by producing type-I IFNs and releasing inflammatory chemokines, such as (C-X-C motif) ligand 10 (CXCL10) and CXCL9, through various mechanisms [5,35]. First, intracellular DAMPS can trigger cancer cell–autonomous signaling to stimulate a type-I IFN response. Indeed, aberrant RNA molecules can be released and detected by the endosomal pattern recognition receptor Toll-like receptor 3 (TLR3) on the cancer cell surface, thus stimulating the production of IFNs in an autocrine and paracrine fashion [35]. Secondly, extra-nuclear DNA in the form of micronuclei and mitochondrial DNA can be sensed by cyclic GMP–AMP synthase (CGAS), thus activating the stimulator of interferon response cGAMP interactor (STING) pathway and the downstream type-I IFN response [5,36,37]. 

While the role of type-I IFNs in the antiviral immune response is well known, they’ve only recently begun to be appreciated in the context of anti-cancer immunity and particularly as a hallmark of effective chemotherapy [38]. IFN production by either cancer cells or tumor-infiltrating dendritic cells promotes autocrine and paracrine signaling that eventually increases immune competence. Among several functions, type-I IFNs promote DC maturation, processing, and antigen presentation to T cells [39]. They act as chemotactic attractants of T cells and promote the functions of effector T cells by (i) increasing their cytotoxic potential through increased expression of perforin 1 and granzyme B [40], (ii) promoting the survival of memory CTLs [41] and (iii) protecting CTLs from natural killer (NK) cell attack [42,43]. Type I IFNs also decrease regulatory T (Treg) cell–mediated immunosuppression by decreasing their activity through upregulating phosphodiesterase 4 and depleting cyclic AMP [44]. 

As such, the therapeutic efficacy of several chemotherapeutics, including bortezomib, benefit greatly from activating the IFN response. The absence of intact IFN signaling hinders the optimal in vivo response, and type-I IFN gene signatures within cancer cells are correlated with better long-term clinical outcomes in different cancers, including MM [5,35,38]. Indeed, transcriptional activation of IFN-stimulated genes, identified within an ICD signature, characterizes patients that perform better clinically after treatment with bortezomib [5]. 

Early immunotherapeutic approaches have clinically used interferon to stimulate immune competence. Even in MM, IFN-alpha demonstrated efficacy as an immune stimulant, although its clinical utility was limited by significant toxicity [5,45]. To overcome this limitation, novel immunotherapeutic approaches, such as STING agonists, are being developed and tested in the clinical setting to stimulate production of IFNs and inflammatory chemokines by cancer cells and immune cells. Preclinical evidence suggests the therapeutic potential of STING agonists in MM, either alone or in combination with bortezomib or other immune therapies [5].

#### 2.1.4. Cell Death and Loss of Plasma Membrane Integrity 

As previously stated, immunogenic potential is not associated with a specific cell death modality but is mainly dependent on the emission of DAMPs [9]. During the late stage of cell death, the loss of plasma membrane integrity promotes tumor immunogenicity by allowing the passive release of “hidden” intracellular proteins that function as DAMPs [46]. However, not all of the released molecules act as proinflammatory stimuli, and their functions are very much disease- and drug-dependent [46]. 

High-mobility group box 1 (HMGB1) protein is an abundant non-histone chromatic protein released after permeabilization of the nuclear lamina and plasma membrane following cell death [47]. Released HMGB1 can function as a proinflammatory stimulus by binding to pattern recognition receptors, specifically TLR4, to stimulate DC maturation and cross-presentation of tumor antigens to T cells [30,48] and by stimulating myeloid differentiation primary response protein-88 to mediate the activation of APCs [49]. In addition, HMGB1 synergizes with ATP to promote NLRP3 inflammasome activation in DCs [28]. HMGB1 can also be released within inflammatory complexes such as those formed with single-stranded DNA, lipopolysaccharide, IL-1β and nucleosomes, which interact with TLR9, TLR4, IL-1R, and TLR2 receptors [46]. 

The ubiquitously expressed cytosolic protein annexin A1 (ANXA1) can also act as a DAMP by facilitating corpse–DC physical interaction [6,50]. Although DC recruitment into the tumor bed is mainly mediated by additional chemokines and ATP, ANXA1 guides DCs towards dying cancer cells by interacting with formyl peptide receptor 1 (FPR1) [50]. The binding facilitates the stable interaction between DCs and cancer cell corpses and the subsequent antigen processing and presentation [50]. 

### 2.2. Perception of Immunogenicity by Immune Cells: A Focus on MM

DAMPs generally stimulate the immune system and cause inflammation by dictating adjuvanticity of cell death. However, effective ICD also requires:(1) antigenicity, or in other words, targetable neo-antigens caused by mutations, genomic instability, and post-translational modifications; and (2) a microenvironment conducive to immune attack (Figure 2) [6,9,51,52,53]. 

MM is a cancer with an intermediate mutational load, where mutational and neoantigen burden increases with disease progression through chromosomal aberrations, aneuploidy, somatic mutations, and DNA damage [54,55,56,57,58]. High mutational and neoantigen load have been associated with poor clinical outcome in MM patients [54,59,60]. This may seem counter-intuitive given the immune properties of neo-antigens, but neo-antigens alone fail to activate an effective adaptive immune response when not associated with sufficient DAMP exposure which ensures adjuvanticity [6,7]. Without DAMPs to provide co-stimulatory signals, antigen presentation to T cells leads to T cell tolerance and anergy [6,7,61,62]. As such, massive tumor antigen release, for example after high-dose (HD) alkylating agents, may have a detrimental effect on the immune system, even though the overall effects are drug- and disease-specific and should be evaluated in a disease context [3]. In MM, transplant-eligible patients undergo HD melphalan before autologous stem-cell transplantation, which has been associated with a higher mutational load (*Samur MK, ASH 2020*). Although studies in other diseases suggest the ability of melphalan to trigger an anti-tumor immune response via ICD, in MM the overall impact of HD melphalan on the immune system is not well understood [63,64].

In a similar scenario, the mere presence of DAMPs after immunogenic stress may only trigger an inflammatory response without engaging adaptive immunity when not associated with sufficient antigenicity [6,7]. An effective immune response following ICD not only requires DAMPs and neoantigens, but also a microenvironment conducive to immune attack. Tumors with higher infiltration of Treg cells, DCs expressing the immunosuppressive enzyme indoleamine 2,3-dioxygenase 1 or an elevated fibrotic response that physically impedes cell circulation are less prone to support the access of efficiently primed CTLs to the tumor niche and the establishment of immunological memory [6,7]. The MM microenvironment is highly immunosuppressive, with early changes in the BM immune composition already detectable at the precursor stages of monoclonal gammopathy of undetermined significance (MGUS) and smoldering MM (SMM) [2,65,66]. Thus, it does not support optimal immune activation. In this scope, the identification of optimal therapeutic combinations and treatment schedules is instrumental to overcoming immune dysfunction and promoting an anti-tumor immune response. This aspect will be discussed in the following sections.

## 3. Challenges of Immunogenic Chemotherapy in Multiple Myeloma

The immunogenic effects of chemotherapy, as well as how cancer cells can become resistant to those immunogenic effects, are relatively understudied in MM. These knowledge gaps represent several challenges for the rational integration of chemotherapy into modern immunotherapy. Here, we will discuss several obstacles that need to be overcome to achieve an effective immune response after immunogenic chemotherapy in the context of MM biology (Figure 3). 

### 3.1. Tumor-Dependent ICD Resistance Mechanisms

Tumor cells can subvert ICD by three main mechanisms. First, they can have a resistance to cell death itself. Resistance to cytotoxicity induced by anti-MM agents has already been described and has informed combination approaches to restore cell death [1]. Second, they can lose antigenicity. Cancer cells may avoid CTL attack via the loss of beta-2-microglobulin or by downregulating major histocompatibility complex class I (MHC-I) expression via epigenetic silencing or loss of heterozygosity at the human leukocyte antigen (HLA) locus [6,7,67,68]. However, in MM, the expression of MHC-I usually increases during progression, although the mechanisms behind this are not clear yet [56,69]. Loss of specific antigens in MM has been instead described as a mechanism of resistance to antigen-targeting monoclonal antibody (mAb) or chimeric-antigen-receptor (CAR) T cells [70,71]. 

Third, the cancer cells can subvert DAMP exposure, leading to an inefficient immune response. Though the knowledge of MM-specific mechanisms for this is still poor, several strategies have been described for other tumors [6]. During the induction of ICD, phagocytosis is strictly dependent on CRT exposure. Therefore, tumor cells can avoid phagocytosis by (1) disrupting the ER stress pathway via reduced eIF2α phosphorylation, (2) secreting mutated forms of CRT (which saturate CD91 receptors on DCs), (3) trapping CRT in the mitochondria in high Stanniocalcin 1 tumors or (4) altering CRT binding sites on the cell surface (asialoglycans) [6,72,73,74,75,76]. Similarly, contextual expression of “don’t eat me” signals, such as CD47, may antagonize CRT and limit phagocytosis [4]. In fact, CD47 blockade has been extensively studied in MM as a promising therapeutic strategy to increase macrophage-mediated phagocytosis and the killing of MM cells [77]. However, little is known about the potential synergistic effect of CD47 inhibition and ICD induction. 

Cancer cells may similarly self-limit the activation of autophagy, thus impairing ATP release [78]. ATP can also be converted into adenosine (highly immunosuppressive) by two ectonucleotidases, CD39 and CD73, so high expression of both enzymes by either cancer cells or immune cells can inactivate ATP [79,80]. In this vein, reduced expression of HMGB1 or ANXA1 protein is associated with poor immune infiltration and minimal DC–corpse interaction [30,81]. Cancer cells can also evade immunogenic response by blocking type-I IFN signaling. In this regard, low levels of STING in MM patient cells correlate with low expression of IFN-stimulated genes (defined as an ICD-signature), which negatively impact the clinical outcome of MM patients treated with ICD inducer BTZ [5]. 

### 3.2. Host-Dependent Mechanisms

Inherited and acquired systemic immune defects can also alter DAMP perception. A loss-of-function Single nucleotide polymorphism in the gene coding for FPR1 (rs867228) is associated with impaired antitumor immunity after ICD induction in breast and colorectal cancer patients. Fpr1-deficient DCs from mice fail to localize to cancer cells succumbing to ICD and have reduced antigen presentation ability [6]. However, how inherited immune defects affect MM progression and treatment have not been studied in a systematic way.

Disease-specific immune alterations represent another challenge. Progressive dysfunction of immune and accessory cells in the bone marrow microenvironment is a feature of MM and leads to loss of immune surveillance, MM cell proliferation, and chemotherapeutic resistance [1,2]. Myeloid-derived suppressor cells, tumor-associated macrophages, mesenchymal stromal cells, plasmacytoid dendritic cells, and osteoclasts contribute to the immune suppression and immune exhaustion [2]. T cell ability to control tumor progression is lost in MM. A decrease in the population of stem-like/resident memory T cells as well as a proliferation of Tregs and Th17 cells contribute to the immune dysfunction and may compromise the efficacy of immunogenic chemotherapy and immunotherapy, particularly T cell–directed therapy [65,82,83,84,85]. Although these immune alterations confer immunosuppression, they similarly represent ideal targets for novel therapeutics.

### 3.3. Drug-Dependent Mechanisms

How MM drugs influence the immune system is understudied. A good first step would be to characterize the side effects of chemotherapeutics on the immune system, particularly by defining the good off-target effects, like those that eventually lead to immune stimulation, versus the bad off-target effects, like immunosuppression. Among the bad off-target effects are myelosuppression and lymphopenia, which can contribute to immunodeficiency and susceptibility to infection in MM patients [86]. 

An enlightening example is that of the glucocorticoids, which constitute the backbone of anti-MM therapy in several treatment regimens [86,87]. Glucocorticoids have direct cytotoxic activity on MM cells and improve chemotherapy-associated nausea and vomiting in patients. However, recent evidence in other cancer types shows glucocorticoids and stress-induced endogenous glucocorticoids may compromise the anti-tumor immune response induced by ICD, specifically by upregulating the expression of Tsc22d3 in DCs and thus hampering their differentiation and antigen presentation [3,6,7,88]. Moreover, glucocorticoids suppress the production of inflammatory chemokines, repress the expression of genes correlated with adaptive immunity or T cell function, and modulate NK activation [3,89,90]. Nevertheless, the overall effects of glucocorticoids seem cell- and disease-specific, and their effect on host immunological competence, alone or in combination with other MM therapies, should be assessed [87,91]. Chemotherapy may also influence whole body physiology and immunological competence by altering gut microbiome composition or modulating stress-related neuroendocrine circuitry [6,7]. This information highlights the need to better define the overall effects of therapeutic agents, either alone or in combination, to inform clinical practice.

## 4. Promises of Immunogenic Chemotherapy in Multiple Myeloma

Although there are several challenges, the optimal integration of immunogenic chemotherapy into modern immunotherapy holds great promise for the treatment of MM patients (Figure 3). Beyond ICD induction, chemotherapy can directly lead to immune activation by acting on immune cells (beneficial off-target effects). Activation of immune effectors such as DCs or CTLs, selective depletion of immunosuppressive cells, and transient lymphodepletion represent just a few of the tumor-extrinsic effects of some clinically employed chemotherapies [6,7,18]. Within MM specifically, the immunomodulatory agent (IMiD) thalidomide and its more potent derivatives lenalidomide and pomalidomide modulate the immune microenvironment through several mechanisms including activating cytotoxic CD8^+^ T, NK, and NKT cells and decreasing the population of Tregs [2,92]. Similarly, the Histone deacetylase 6 -specific inhibitor ACY241 exerts its anti-MM activity, at least in part, by enhancing the anti-tumor response of antigen-specific central memory CTLs [2,93]. Finally, proteasome inhibitors can also disrupt the interaction of MM cells with the bone marrow milieu by decreasing the expression of adhesion molecules, inhibiting angiogenesis, and modifying osteoclast activity and bone turnover [2,94]. 

In the context of ICD, investigating tumor resistance mechanisms may inform novel combination approaches to overcome ICD resistance. Restoration of cell death, proper DAMP exposure, and immune perception of ICD are necessary for adaptive recognition of cancer cells. ER stressors, such as thapsigargin or eIF2α phosphatase inhibition, have shown promise for inducing and restoring the ISR to overcome insufficient CRT exposure [6,72,95,96]. Similarly, blockade of “don’t eat me” signals, such as CD47, can inhibit endogenous molecules that suppress ICD-mediated adaptive immunity [6]. Blockade of CD47 via mAb has been extensively studied in MM as a promising therapeutic strategy to increase macrophage-mediated phagocytosis and the killing of MM cells [77]. However, little is known about the potential synergistic effect of CD47 inhibition and ICD induction in MM. 

Autophagy stimulation offers another viable avenue of investigation. Autophagy inducers increase ATP secretion, and mAb targeting CD39 or CD73 can overcome ATP degradation in other cancers [26,31,80]. Treatment with TLR3 ligand polyinosinic:polycytidylic acid (polyI:C) overcomes the defective perception of immunogenicity in ANXA1- or FPR1-deficient conditions [97]. Additional strategies can be employed to reinstate ICD-induced type-I IFN response. Stimulation of PRRs, such as TLR3 or STING agonist, as well as recombinant type-I IFN have successfully rescued pro-inflammatory IFN expression [6]. The activation of the STING pathway is an emerging immunotherapeutic approach in several cancers under evaluation in the context of clinical trials [98]. Recent evidence suggests that STING agonists may also represent a promising therapeutic approach in MM, and that their combined use with BTZ may increase immune activation specifically in patients with a low basal level of STING expression and poor type-I IFN response [5].

Lastly, immunogenic chemotherapy is the ideal partner for immunotherapy, as inducing ICD can transform an immunogenic “cold” microenvironment into a “hot” one. By increasing the immune infiltration of the tumor bed, ICD inducers adapt the microenvironment for successful immune-based therapies. Several types of immunotherapies have already been tested and, in some cases, translated into the clinical management of MM patients [2]. With the availability of mAb, Antibody Drug Conjugate, Bi-Specific Antibodies, CAR-T cells, immune checkpoint inhibitors (CPIs), and vaccine strategies, it is imperative to identify the optimal combination, treatment setting, and sequence of therapies to improve long-term response. Already, several clinical trials are testing the benefit of combining ICD inducers with CPIs to increase the portion of patients responding to CPIs. Several excellent reviews cover this strategy in detail [7]. Despite promising preclinical data, early clinical trials of PD-1/PD-L1 blockade in MM therapy have been discouraging, with immune-related toxicities in combination with IMiDs [1,2,99,100]. In light of these data, one could speculate that future analysis of combinations of ICD inducers and CPIs may increase therapeutic response. Similarly, cataloging the immunogenic potential of anti-MM agents like BTZ may inform their clinical use in combination with immune therapies and reveal the mechanisms underlying their synergistic activity with other agents (such as IMiDs) in patients. Moreover, understanding the ideal timing of ICD/immunotherapy combinations (i.e., during the precursor disease when the immune system is not fully dysfunctional, or after transplant when immunosuppressive cells are consistently decreased in population) is imperative for effective and durable immune activation.

## 5. Conclusions

Awareness of the immunogenic potential of chemotherapy has transformed the treatment paradigm of several cancers and will improve MM treatment as well. A deeper understating of immune dysfunction in MM has already led to the development of an effective immune-based approach. In this scenario, it is therefore essential to identify combination therapeutic regimens in which intrinsic and extrinsic immunochemotherapeutic effects can synergize to optimize immune system activation and overcome immune escape. Combining ICD inducers with immunotherapy holds great promise for mounting an effective long-lasting tumor-specific immune response in MM patients. 

## Figures and Tables

**Figure 1 cells-11-02519-f001:**
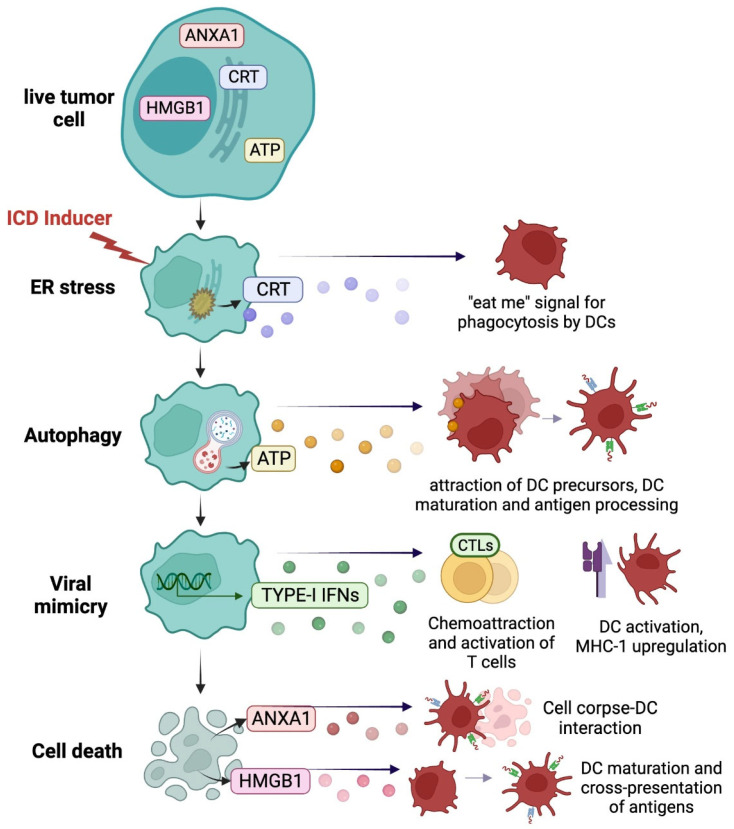
**Immunogenic DAMPs from dying cancer cells.** ICD inducers stimulate a series of stress and death pathways in malignant cells, which leads to the exposure of immunogenic DAMPs. In response to ER stress, cells translocate intracellular CRT to the cell surface where it acts as a potent “eat me” signal, promoting cell engulfment and phagocytic clearance by DCs. The release of ATP following autophagy promotes the attraction and maturation of DC precursors and antigen processing. ICD-stressed cells also enter a viral mimicry state, which results in the release of type I IFNs that promote T cell attraction and activation, and DC activation. Furthermore, loss of plasma membrane integrity leads to the release of “hidden molecules,” including HMGB1 and ANXA1, that stimulate cross-presentation of tumor antigens by DCs to T cells and facilitate DC–corpse interaction. *Abbreviations: ANXA1, annexin A1; CRT, calreticulin; DC, dendritic cell; ER, endoplasmic reticulum; HMGB1, high mobility group box 1; IFN, interferon; ICD, immunogenic cell death; CTL, cytotoxic T lymphocytes*.

**Figure 2 cells-11-02519-f002:**
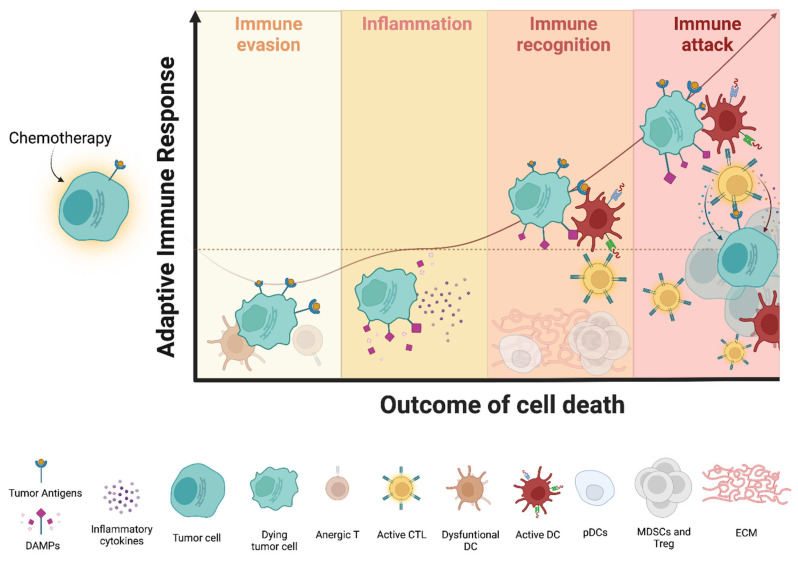
**The outcome of cell death on adaptive immunity.** The perception of chemotherapy-induced cell death by the immune system and the extent of the activation of adaptive immunity rely on the coexistence of antigenicity, adjuvanticity, and a permissive microenvironment. During cell death, four different scenarios can be hypothesized: *(**i)* immunogenic neo-antigens with poor co-stimulatory signals and poor adjuvanticity lead to T cell tolerance and anergy, thus favoring immune escape; *(ii**)* adjuvanticity (provided by DAMPs) associated with insufficient antigenicity may trigger only cell death-induced “inflammation”; *(**iii)* antigenicity and adjuvanticity together can effectively stimulate adaptive T cell recognition of cancer cells, although an immunosuppressive microenvironment can still limit anti-tumor attack; *(iv**)* the presence of a permissive microenvironment amplifies immune attack by allowing for immune infiltration and T cell–mediated killing. *Abbreviations: DAMPs, damage-associated molecular patterns; CTL, cytotoxic T lymphocytes; DC, dendritic cell; pDC, plasmacytoid DC; MDSCs, myeloid-derived suppressor cells; Treg, regulatory T cells; ECM, extracellular matrix*.

**Figure 3 cells-11-02519-f003:**
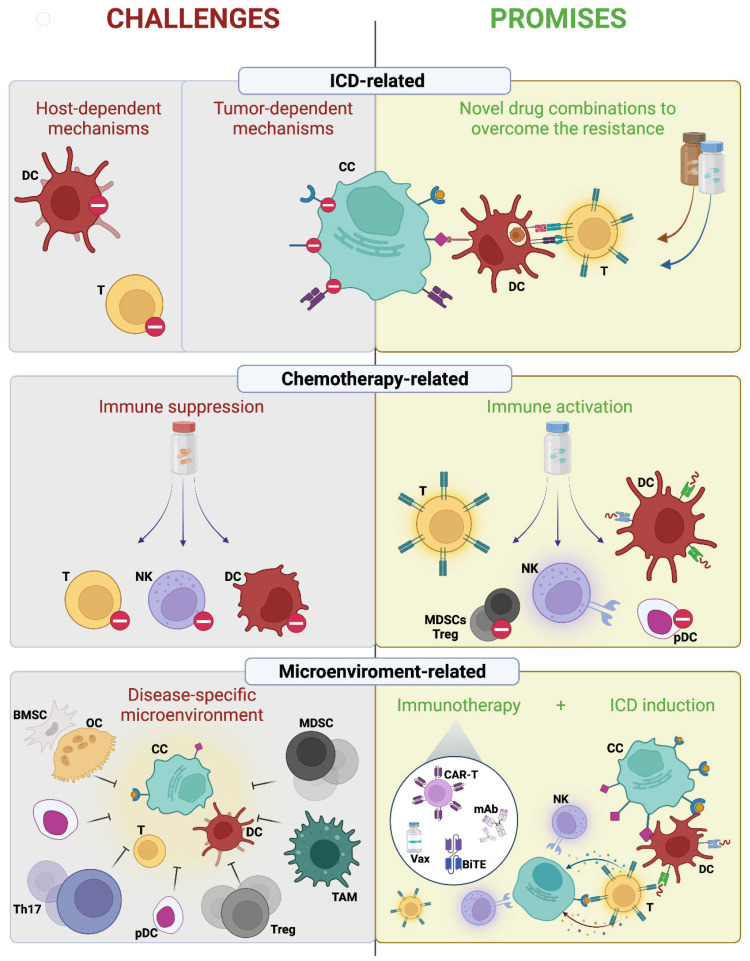
**The challenges and promises of immunogenic chemotherapy in multiple myeloma.** Several challenges limit the successful induction of an anti-tumor immune response; however, they may be overcome and thus represent novel opportunities. First, ICD resistance dictated by various host- or tumor-dependent mechanisms poses a significant challenge. However, the identification of novel drug combinations that will restore cell death, proper DAMP exposure, and immune perception of ICD show promise for overcoming resistance and restoring immunosurveillance. Second, chemotherapy may alter immune composition towards immune suppression. Conversely, several chemotherapeutics can promote immune activation by directly acting on immune effector cells. Thus, the assessment of drug-specific effects within the disease context may inform the preferential use of immune-activating agents. Finally, the disease-specific immunosuppressive BM-microenvironment may limit a long-lasting adaptive immune response. The optimal integration of ICD inducers into modern immunotherapy has the potential to promote immune attack and long-term memory immune recognition to convert a “cold” tumor microenvironment into a “hot” one. *Abbreviations: DC, dendritic cell; CTL, cytotoxic lymphocyte; NK, natural killer cell; Treg, regulatory T cell; MDSC, myeloid-derived suppressor cell; pDC, plasmacytoid dendritic cell; Th17, T Helper 17 cell; BM SC, bone marrow stromal cell; TAM, tumor-associated macrophage; OC, osteoclasts; CC, cancer cell; CAR, chimeric antigen receptor; mAb, monoclonal antibody; Vax, vaccine; BiTE, bispecific T cell engager*.

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
