# Peer review of "Promises and Challenges of Immunogenic Chemotherapy in Multiple Myeloma"

_cells, 2022, doi:10.3390/cells11162519_

Round 1

Reviewer 1 Report

In the manuscript, cells-1843481 by Johnstone et al, authors have reviewed the literature to emphasize the promises and challenges of immunogenic chemotherapy in multiple myeloma. Given that the approaches being studied and used for the treatment of multiple myeloma include the killing of tumor cells directly through anti-tumor agents and immunotherapy, and indirectly by activating the immune system, it is pertinent to explore and understand their integration. It would add clinical benefit in terms of developing efficient treatment options. Subject of this review article is quite interesting, and the manuscript is comprehensive. However, there are certain points to be considered to improve.  

Major-

1.    Manuscript organization appears very casual. Sections, heading, subheading and paragraphs need some revision for coherence. Heading and subheadings are not clear in the current manuscript file. Some are italics and some are regular. It becomes confusing to get which one is heading or which one is subheading.

2.    In some headings there are many small paragraphs (2–3-line paragraphs) that do not make add anything to the message rather break the readability. Most of those paragraphs do not end with a concluding message. Please organize the heading, subheading properly. Please consider trying better choice words for some subheadings to have coherent message throughout.

3.    Is there any specific reason, authors have provided the “promises” after the “challenges” in immunogenic chemotherapy? I would think of starting with “promises” first and then talk about “challenges” later on.

4.    Abstract and Introduction sections do not discuss much about multiple myeloma. In other words, why multiple myeloma was of the interest for this review article?

Minor-

1.    The third paragraph in the “Introduction” section duplicated/repeated. Please remove the following repeated paragraph “In this review, we will summarize the present knowledge on the mechanisms under-lying ICD and how we can exploit it in the current MM treatment landscape. Furthermore, we will discuss the challenges of immune evasion and ICD resistance. We anticipate that the integration of such knowledge will improve chemo- and immunotherapy combinations that will impact MM treatment and improve the long-term survival of MM patients.”

2.    Immunogenic cell death concept is not well explained.

3.    Second paragraph of the subheading “Immunogenic cell death: using cancer to beat cancer”, Is this is heading? “Immunogenic danger signals from dying cancer cells”. In the same paragraph, last sentence, “The successful induction of ICD is defined by the quality of the cellular dialog be-tween dying cancer cells and the immune system [6,7,9].” is not clear. Please revise with better choice of words.

4.    In the following paragraph, last two sentences “Here, we will describe the different signaling within the scope of MM biology. Therefore, there are multiple points at which tumor cells or therapeutic agents can amplify or minimize the impact of ICD.” What does "Therefore" imply here?? Please revise these two sentences for coherence.

5.    In subheading “ER stress response and CRT exposure “.  Please consider defining CRT somewhere in the main text before using abbreviation in a heading. It is defined in Figure 1 legend only. 

6.    Second line of the above-mentioned heading. “Therefore, they are particularly susceptible to therapeutic interventions that disequilibrate protein homeostasis such as proteasome inhibitors like bortezomib [5,10].” Please revise the sentence for better readability. Issue is "therapeutic interventions....... such as proteasome inhibitors like bortezomib"

7.    Another sentence in the same heading “The ISR pathway involves activating eukaryotic translation initiation factor 2 subunit alpha (eIF2α), and this phosphorylation is a hallmark of effective ICD induction.” “this phosphorylation” as in? Please consider revising the first segment of the sentence accordingly.

8.    Heading “2). Autophagy and ATP release” has 5 paragraphs. The logic of making extra paragraph of 2-3 lines is not clear. moreover, it breaks the readability.

9.    It would be helpful if authors revise the manuscript thoroughly for such minor issues.

10. Please change the reference fonts according to the journal guidelines.  

Author Response

Point-by-point reply

We are very grateful to all Reviewers for their thoughtful comments that helped us to significantly improved our manuscript. We have revised the manuscript organization to better emphasize the relevance of immunogenic chemotherapy in multiple myeloma.

Reviewer #1:

In the manuscript, cells-1843481 by Johnstone et al, authors have reviewed the literature to emphasize the promises and challenges of immunogenic chemotherapy in multiple myeloma. Given that the approaches being studied and used for the treatment of multiple myeloma include the killing of tumor cells directly through anti-tumor agents and immunotherapy, and indirectly by activating the immune system, it is pertinent to explore and understand their integration. It would add clinical benefit in terms of developing efficient treatment options. Subject of this review article is quite interesting, and the manuscript is comprehensive. However, there are certain points to be considered to improve. 

Major

  1. Manuscript organization appears very casual. Sections, heading, subheading and paragraphs need some revision for coherence. Heading and subheadings are not clear in the current manuscript file. Some are italics and some are regular. It becomes confusing to get which one is heading or which one is subheading.

We thank the reviewer for this important comment. As suggested, the manuscript organization and formatting have been revised. Numbers have been included in the paragraphs to facilitate readability and improve clarity.

  1. In some headings there are many small paragraphs (2–3-line paragraphs) that do not make add anything to the message rather break the readability. Most of those paragraphs do not end with a concluding message. Please organize the heading, subheading properly. Please consider trying better choice words for some subheadings to have coherent message throughout.

We thank the reviewer for this important point. As suggested, the paragraphs within the headings have been revised to improve readability and cohesion with the heading and subheadings (lines 58-66; 288-290).

  1. Is there any specific reason, authors have provided the “promises” after the “challenges” in immunogenic chemotherapy? I would think of starting with “promises” first and then talk about “challenges” later on.

We thank the reviewer for this comment. We discussed the challenges of immunogenic chemotherapy in MM first to allow the readers to better understand the strategies to overcome them, as discussed in the “Promises” section. Moreover, the discussion of the promises was more in line with the concluding remarks.

  1. Abstract and Introduction sections do not discuss much about multiple myeloma. In other words, why multiple myeloma was of the interest for this review article?

This is a relevant point and we thank the reviewer for this comment. As suggested, abstract and introduction have been revised accordingly (lines 21-23; 38-41; 48-49).

Minor

  1. The third paragraph in the “Introduction” section duplicated/repeated. Please remove the following repeated paragraph “In this review, we will summarize the present knowledge on the mechanisms under-lying ICD and how we can exploit it in the current MM treatment landscape. Furthermore, we will discuss the challenges of immune evasion and ICD resistance. We anticipate that the integration of such knowledge will improve chemo- and immunotherapy combinations that will impact MM treatment and improve the long-term survival of MM patients.”

The repeated paragraph has been deleted (lines 52-56).

  1. Immunogenic cell death concept is not well explained.

We thank the reviewer for this clarification. The “Immunogenic cell death: using cancer to beat cancer” paragraph has been revised to better state immunogenic cell death concept (lines 58-66).

  1. Second paragraph of the subheading “Immunogenic cell death: using cancer to beat cancer”, Is this is heading? “Immunogenic danger signals from dying cancer cells”. In the same paragraph, last sentence, “The successful induction of ICD is defined by the quality of the cellular dialog be-tween dying cancer cells and the immune system [6,7,9].” is not clear. Please revise with better choice of words.

We thank the reviewer for this comment. “Immunogenic danger signals from dying cancer cells” is the subheading and the manuscript organization has been revised as detailed above. As suggested, the last sentence of the paragraph has been revised for clarity (lines 74-76).

  1. In the following paragraph, last two sentences “Here, we will describe the different signaling within the scope of MM biology. Therefore, there are multiple points at which tumor cells or therapeutic agents can amplify or minimize the impact of ICD.” What does "Therefore" imply here?? Please revise these two sentences for coherence.

We thank the reviewer for this comment. The word “Therefore” has been deleted and the sentences edited for clarity (lines 81-84).

  1. In subheading “ER stress response and CRT exposure “. Please consider defining CRT somewhere in the main text before using abbreviation in a heading. It is defined in Figure 1 legend only.

We thank the reviewer for this comment. The heading has been revised and the abbreviation provided in the main text (lines 98 and 108).

  1. Second line of the above-mentioned heading. “Therefore, they are particularly susceptible to therapeutic interventions that disequilibrate protein homeostasis such as proteasome inhibitors like bortezomib [5,10].” Please revise the sentence for better readability. Issue is "therapeutic interventions....... such as proteasome inhibitors like bortezomib"

We thank the reviewer for this clarification. The sentence has been revised accordingly (lines 100-102).

  1. Another sentence in the same heading “The ISR pathway involves activating eukaryotic translation initiation factor 2 subunit alpha (eIF2α), and this phosphorylation is a hallmark of effective ICD induction.” “this phosphorylation” as in? Please consider revising the first segment of the sentence accordingly.

We thank the reviewer for this clarification. The sentence has been revised accordingly (lines 105-107).

  1. Heading “2). Autophagy and ATP release” has 5 paragraphs. The logic of making extra paragraph of 2-3 lines is not clear. moreover, it breaks the readability.

We thank the reviewer for this clarification. The paragraphs have been combined to improve readability.

  1. It would be helpful if authors revise the manuscript thoroughly for such minor issues.

We thank the reviewer for this point. Minor issues have been revised throughout the manuscript.

  1. Please change the reference fonts according to the journal guidelines.

Reference fonts have been modified according to the MDPI Reference List and Citations Style Guide.

Reviewer 2 Report

I read the manuscript “Promises and challenges of immunogenic chemotherapy in multiple myeloma”. I found it interesting with an important clinical significance.

Comments to the authors are as follows:

- Please remove the abbreviations that are not necessary and introduce the missing abbreviations. 

- In the text, there are typing mistakes and all layout issues should be resolved to submit a neat paper. The authors should carry out an important and careful improvement.

Author Response

Point-by-point reply

We are very grateful to all Reviewers for their thoughtful comments that helped us to significantly improved our manuscript. We have revised the manuscript organization to better emphasize the relevance of immunogenic chemotherapy in multiple myeloma.

Reviewer #2:

I read the manuscript “Promises and challenges of immunogenic chemotherapy in multiple myeloma”. I found it interesting with an important clinical significance.

Comments to the authors are as follows:

- Please remove the abbreviations that are not necessary and introduce the missing abbreviations.

We thank the reviewer for this suggestion. Unnecessary abbreviations have been removed and missing abbreviations have been revised accordingly.

- In the text, there are typing mistakes and all layout issues should be resolved to submit a neat paper. The authors should carry out an important and careful improvement.

We thank the reviewer for this comment. Manuscript layout, typos and other minor issues have been carefully revised throughout the manuscript.